# 3D-Printed Polyester-Based Prototypes for Cosmetic Applications—Future Directions at the Forensic Engineering of Advanced Polymeric Materials

**DOI:** 10.3390/ma12060994

**Published:** 2019-03-26

**Authors:** Joanna Rydz, Wanda Sikorska, Marta Musioł, Henryk Janeczek, Jakub Włodarczyk, Marlena Misiurska-Marczak, Justyna Łęczycka, Marek Kowalczuk

**Affiliations:** 1Centre of Polymer and Carbon Materials, Polish Academy of Sciences, 34, M. Curie-Skłodowska St., 41-800 Zabrze, Poland; wsikorska@cmpw-pan.edu.pl (W.S.); mmusiol@cmpw-pan.edu.pl (M.M.); hjaneczek@cmpw-pan.edu.pl (H.J.); jwlodarczyk@cmpw-pan.edu.pl (J.W.); 2Laboratorium Kosmetyczne Dr Irena Eris Sp. z o.o., R&D Department, 12 Armii Krajowej St., 05-500 Piaseczno, Poland; marlena.misiurska-marczak@drirenaeris.com (M.M.-M.); justyna.leczycka@drirenaeris.com (J.Ł.); 3Faculty of Science and Engineering, University of Wolverhampton, Wulfruna Street, Wolverhampton WV1 1SB, UK

**Keywords:** three-dimensional printing, (bio)degradable polyester, prototype container, compostability, biodegradability, weathering test

## Abstract

Knowledge of degradation and impairment phenomena of (bio)degradable polymeric materials under operating conditions, and thus the selection of test procedures and prediction of their behavior designates the scope and capabilities as well as possible limitations of both: the preparation of the final product and its durability. The main novelty and objective of this research was to determine the degradation pathways during testing of polylactide and polylactide/polyhydroxyalkanoate materials made with three-dimensional printing and the development of a new strategy for the comprehensive characterization of such complex systems including behavior during waste disposal. Prototype objects were subjected to tests for damage evolution performed under simulating operating conditions. The reference samples and the tested items were characterized by gel permeation chromatography and differential scanning calorimetry to determine changes in material properties. The studies showed that: polyhydroxyalkanoate component during accelerated aging and degradation in environments rich in microorganisms accelerated the degradation of the material; paraffin accelerates polylactide degradation and slows degradation of polyhydroxyalkanoate-based material; under the influence of an environment rich in enzymes, paraffin contamination accelerates biodegradation; under the influence of natural conditions, paraffin contamination slowed degradation; the processing conditions, in particular the printing orientation of individual parts of the container, influenced the material properties in its various regions, affecting the rate of degradation of individual parts.

## 1. Introduction

The use of environmentally friendly polymers for long shelf-life applications as cosmetic packages is a very important trend that should be developed. Materials intended for contact with cosmetic products must perform not only certain functions, but also must meet acceptable standards of safety during use and exhibit chemical and physical stability during storage or transportation. Not only information about cosmetic ingredients, their toxicological profile, traces, or additives that it may contain should be known but also the information on the packaging material, in particular its purity and stability is needed [1]. Therefore, it is important to identify the interactions between polymer container and the cosmetic formulation that it contains [2,3]. In this case, the complete replacement of conventional plastics with environmental-friendly packaging is impossible to achieve without recognizing and assessing the possible risks [4,5].

During the preliminary study, polylactide (PLA) limitations have already been observed for applications in cosmetics and household chemistry packaging. Recently, it has been revealed that the process of PLA films degradation occurs in a hydrophobic solvent, such as paraffin, due to the residual moisture content [2,3,6]. Despite of some advantageous properties, pure PLA is also stiff and fragile near its glass transition temperature (*T_g_* in the range from 55 to 65 °C), which limits its applications at ambient temperature. In order to expand the range of applications of (bio)degradable polymeric materials containing polylactide, various methods are used to modify PLA properties depending on the intended application, such as mixing with other polymers or additives as natural fillers to obtain blends and composites or copolymerization reaction of lactide with other cyclic esters to obtain (bio)degradable copolymers [7]. An interesting alternative to conventional plastic packaging for cosmetics industry may be use of polyhydroxyalkanoates (PHA)s as modifiers of PLA materials [8,9,10,11]. Both PLA and PHA are thermoplastics that are used for traditional extrusion and can also be used in the three-dimensional (3D) printing process, hence (bio)degradable polymers open up new possibilities for 3D technology [12].

3D printing, a manufacturing technology that is able to create 3D objects from a computer-aided design (CAD) file by deposition of multiple layers of material, offers significant advantages over other traditional subtractive manufacturing techniques that allow creating spatial models by removing material using cutting tools. The main advantage is the possibility of produce elements with a designed flexible structure using the number of different materials; therefore, it is important for the final products. 3D printing was originally used for prototyping, but recent progress in this area offer for this technology the possibility to be used on a large scale. So far, the precise 3D printing has found application in the fields of biomedical devices and biocompatible structural components [13]. The next direction is capability to design complexed and well-defined complex structures that are difficult to achieve under currently established manufacturing techniques. Moreover, it is increasingly being used in other areas such as automotive and transportation industry, construction, electronic, and textile sectors as well as consumer goods sector (furniture, suitcases, or safety helmets) [12,13].

The use of (bio)degradable and/or bio-based polyesters increases in areas such as disposable products and short-term and long-term packaging applications, and prediction of the materials applications depend on a full understanding of their behavior and performance throughout their life cycle under the real conditions in which the materials will be used and disposed [14,15,16]. The packaging of most cosmetics contains hard-to-remove oily substances that may interfere with conventional recycling. It is therefore necessary to look for new materials that could be used as cosmetics packaging and would not create a threat to the environment. Cosmetics packaging from biodegradable polymers can be subjected to the organic recycling process without removing the residue of their content. On the market, we can find applications of biodegradable packaging for cosmetic products—such as lipsticks, powders, bronzers, or eyeshadows—but it is still a small area of application [17,18,19]. Cosmetics are in contact with the human body (skin, hair, nails) to provide specific functions. Their formulations are therefore complex and to avoid modification of cosmetic properties, the cosmetic composition must be chemically and physically stable during storage and transport. Contaminants can be microbial or chemical and can be caused by improper storage, decomposition reactions, or migration of the products. In general, for packaging from biodegradable and/or compostable polymeric materials, typical end-of-life options (depending on the properties of the polymer) include recycling, monomer recovery (e.g., recovery of lactic acid from PLA and then subsequent use of the recovered monomer), incineration with energy recovery, as well as organic recycling (composting) or landfill disposal [20]. The purpose of composting is the disposal of biodegradable waste and, as a result, to produce a stabilized product that can be used or stored without further treatment. The organic recycling of biodegradable polymeric materials enables controlled biological decomposition of these polymers to safe and eco-friendly products for the environment and human life and health [21,22,23,24].

So far, there are no available standard testing procedures for biodegradable packages of cosmetic products. Based on the knowledge of the cosmetic formulation and packaging material, as well as on the basis of available standards for packaging of medical, pharmaceutical and food products, particular on (i) Guidelines on stability testing of cosmetics product, Colipa 2004 [25]; (ii) EN 1186, Materials and articles in contact with foodstuffs—Plastics [26]; (iii) EU legislation [27,28,29] can be developed an evaluation protocol for the assessment of long-term cosmetic packaging from biodegradable polymers. Therefore, the aim of those studies was to elaborate a new strategy for predicting the behavior of such advanced polymer systems based on the comprehensive characterization of degradation pathways.

## 2. Materials and Methods

### 2.1. Materials

The materials used in this study were commercial PLA (Orbi-Tech, Leichlingen, Germany) and PLA/PHA (ColorFabb, Belfeld, The Netherlands) 3D printing filaments. The total carbon content in the PLA and PLA/PHA samples was 49% and 50%, respectively (determined by elemental analysis performed by means of a Vario EL III apparatus, Elementar, Langenselbold, Germany). The detailed material characteristic was described in [12].

The following simulants were used for the accelerated aging test: deionized water (pH = 5.2, used as reference), liquid paraffin (99.98%, water content: 0.016% by Karl Fischer method and the total carbon content: 88%, from Pharmaceutical Laboratory COEL, Kraków, Poland and anhydrous 99.8% ethyl alcohol (ethanol, water content: 0.035% by Karl Fischer method) from POCH SA, Gliwice, Poland were used without further purification.

### 2.2. Fabrication of 3D-printed Prototypes of Cosmetic Containers

PLA and PLA/PHA prototype of cosmetic containers in the form of jars were obtained using fused deposition modeling printer (FLASHFORGE Dreamer dual extrusion 3D printer, FlashForge Corporation, Jinhua, China) with the methodology of 3D structures creation based on bottom-up layer-by-layer manufacturing directly from a CAD model. The printer settings used are shown in Table 1.

Acetone was used to clean the building platform of any residue after processing the samples. The bottom of the jars and the top of the lids were obtained with flat, XY plane processing build orientation and with raster angle (0°/90°) (cross pattern). The wall of the jar was obtained by printing individual layers, consisting of two concentric, adjoining rings, in the plane XZ one above the other. Cosmetic containers had an average mass of 1.65 g, in particular jars of 0.92 g and lids of 0.73 g for PLA material and 1.72 g, in particular jars of 0.92 g and lids of 0.80 g for PLA/PHA material.

### 2.3. Modeling and Simulation of Degradation and Damage Phenomena

#### 2.3.1. Accelerated Aging Test

An accelerated shelf life study is based on the Arrhenius model and states that the increasing temperature of 10 °C doubles the chemical reaction rate [30,31]. The accelerated aging time (AAT) needed to establish equivalence of aging in real time is determined by dividing the desired (or required) shelf life by an accelerated aging factor (AAF). The AAF is calculated using the following equation:
AAF = *Q_10_*^[(*T*^*_AA_*^− *T*^*_RT_*^)/10]^(1)
where *Q_10_* is an aging factor for 10 °C increase or decrease in temperature, *T_AA_* is accelerated aging temperature (°C), and *T_RT_* is ambient (warehouse) temperature (°C). Using the Arrhenius equation with *Q_10_* equal to 2 is a common and conservative way of calculating an aging factor, especially in the case of medical devices. Accordingly, the selected parameters for the experiment were: *T_AA_* = 55 °C (test temperature is generally ranges from 50 °C to 60 °C, most often 55 °C); *T_RT_* = 22 °C (ambient/storage temperature is typical between 22 °C and 25 °C; 22 °C results in the shortest duration of the test); *Q_10_* = 2. Relative humidity *RH* = 5% is not a factor of the Arrhenius equation, however, the relative humidity should be kept below 20% so in order not to damage the material. For the accelerated aging test PLA and PLA/PHA cosmetic containers (jars with lids) were dried under vacuum at ambient temperature to a constant mass to eliminate ultimate water content and then were filled with 1 mL of a cosmetic simulant (deionized water, paraffin, or ethanol) and incubated at *T_AA_* = 55 °C ± 1 °C together with blank test (empty cosmetic container). After a predetermined accelerated aging times (9, 19, 37, 74, 111, and 185 days corresponding to real-time aging 3, 6, 12, 24, 36, and 60 months respectively) the cosmetic containers were opened and separated from the simulants. For the deionized water and ethanol, the samples were washed with deionized water to remove degradation products and dried under vacuum at ambient temperature to a constant mass. For the viscous simulant, paraffin, the samples were drained on a filter paper. The experiments were performed in triplicate.

#### 2.3.2. Packaging/Cosmetic Formulation Compatibility Test

Tests with real cosmetic formulations were conducted under two temperature conditions, at ambient temperature (23 °C ± 2 °C, control experiment) and in a heating chamber at 45 °C (accelerated aging) for a period of 84 days (12 weeks). In the middle of the test, after 42 days (6 weeks, preliminary assessment), and finally after 84 days (12 weeks, final assessment) the stability of the cosmetic formulations and cosmetic containers was checked. Five different cosmetic formulations (oil-in-water emulsions) ranging from the lightest (moisturizing) to heaviest (oily) in an amount of 1.5 g were selected for the experiment and placed in closed PLA and PLA/PHA cosmetic containers. The experiment was performed in quintuplicate.

#### 2.3.3. Composting under Laboratory Conditions

The biodegradation under laboratory conditions was performed using S/N 110315 Micro-Oxymax respirometer (COLUMBUS INSTRUMENTS, Columbus, OH, USA) equipped with a computer as a controller and a device for recording, archiving and presenting data. Micro-Oxymax respirometer performs periodic measurements in a closed system, which means that the air in the measuring chamber is pumped by a gas sensor and returns to the chamber. The system, work in standard conditions, automatically compensates for changes in pressure and temperature [32]. The installed software allows continuous monitoring of the level of carbon dioxide (CO_2_) emission in a closed system. The tested samples and additionally, the 500 g of thin-layer chromatography analytical grade cellulose as positive control was placed in a glass containers. For control experiment, 500 g of mature compost without sample was placed in a glass container (blank inoculum). The amount of released CO_2_ produced from the tested samples (corrected for that derived from the blank inoculum) was monitored as a function of time. The cumulative value of CO_2_ obtained in the test was used to determine the degree of biodegradation of studied materials. The degree of biodegradation (*B*) was calculated as the ratio of the measured amount of CO_2_ evolved to the theoretical amount of CO_2_ (*Th_CO_2__*):*B* = [(*m_CO_2_test_* − *m_CO_2_blank_*)/*Th_CO_2__*] × 100%(2)
where *m_CO_2_test_* and *m_CO_2_blank_* are the amount of CO_2_ [mg] evolved from tested material and blank inoculum, respectively. The *Th_CO_2__* is calculated based on the total carbon content of the polymer (TC%) assuming the total oxidation of the carbon to CO_2_ according to the equation:*Th_CO_2__* = (44.0095/12.0107) × [(*m_s_* × TC%)/100](3)
where *m_s_* is the amount of sample tested (mg), 44.0095 and 12.0107 molar mass of CO_2_ and atomic mass of C, respectively [33,34].

For the biodegradation test the samples: (i) empty cosmetic containers and (ii) cosmetic containers with a small amount of paraffin (0.3 g) simulating the packaging with cosmetic contamination, three items of tested cosmetic containers were placed in a glass container containing 500 g of mature compost (from the Sorting and Composting Plant in Zabrze, Poland) used as a solid bed, a source of nutrients and an inoculum rich in thermophilic microorganisms. Mature compost used cannot be sterilized as its natural flora (bacteria and fungi together) is necessary to carry out a biodegradation test. The compost was sieved, and the obvious plant and other inert materials were removed. The environmental humidity was 43% for PLA and 42% for PLA/PHA cosmetic containers and incubated time 12 weeks (84 days, according to USA standard ASTM D 6400 [35]) in the average temperature 58 °C. After incubation the samples were removed, cleaned of compost and analyzed.

#### 2.3.4. Degradation under Industrial Composting Conditions

The samples were incubated in cages in BIODEGMA system and in the static composting open-air pile (location: latitude 50° 18’ 30.71” N and longitude 18° 48’ 18.52” E) at the Sorting and Composting Plant in Zabrze, Poland as described in [36]. The BIODEGMA composting system features simple, economic and efficient technological solutions for biodegradation of organic waste, including separately collected biogenic and garden waste, sewage sludge, and the organic component of mixed municipal waste. In the BIODEGMA system, the samples were incubated for 21 days (3 weeks) at an average temperature of 60 °C ± 5 °C with an average pH of 7.5. In the composting pile, samples were incubated for 21 and 84 days (3 and 12 weeks) at an average temperature of 61 °C ± 5 °C with an average pH of 6.9. The cages with the materials studied were placed into composting systems at a depth of 0.5 meter below the compost surface. The samples were run in triplicate and the experiments were conducted between July and October 2018.

#### 2.3.5. Weathering Test

For degradation under natural weathering conditions the PLA and PLA/PHA cosmetic containers were placed on the testing site at the latitude 50°18’52.68” N and longitude 18°46’16.20” E. Experiments were conducted between September 2016 and August 2017 (12 months), with daily rainfall below 43.8 mm (http://www.wunderground.com/) The average open-air temperature was 13 °C.

### 2.4. Characterization

The following, most representative, characterization methods were selected [12,37].

#### 2.4.1. Humidity Analysis

For humidity analysis the AXIS AGS50 (Gdańsk, Poland) moisture analyzer with the halogen heaters was used. The halogen heaters with a nominal power of 200 W and l = 118 mm are connected in series. The material humidity was determinate upon the basis of mass loss during drying the sample (thermo-gravimetrical method).

#### 2.4.2. Gel Permeation Chromatography (GPC) Analysis

The molar mass and molar-mass dispersity of the samples were determined using gel permeation chromatography conducted in chloroform solution at 35 °C with an eluent flow rate of 1 mL/min using a Viscotek VE 1122 (Malvern, UK) pump with two Mixed C PLgel styragel columns (Agilent, Santa Clara, CA, USA) in series and a Shodex SE 61 RI detector (Showa Denko, Munich, Germany). Polystyrene standards (Calibration Kit S-M-10, Polymer Laboratories) with narrow molar-mass dispersity were used to generate a universal calibration curve. The samples were measured using OmniSEC 5.0 (Viscotek, Malvern, UK) software. The molar mass loss was calculated using the equation:% molar mass loss = [(*M_w*0*_* − *M_wx_*)/*M_w*0*_*] × 100%(4)
where *M_w*0*_* is the initial mass-average molar mass and *M_wx_* is the consecutive or final average molar mass.

#### 2.4.3. Differential Scanning Calorimetry (DSC)

Thermal characteristics of the materials were obtained using the TA-DSC Q2000 apparatus (TA Instruments, Newcastle, DE, USA). The instrument was calibrated with high purity indium. The first calorimetric trace (first heating run) of the initial sample in which the thermal history is suppressed was acquired from −30 °C to 220 °C at the heating rate of 20 °C/min and of 5 °C/min, the second and third calorimetric traces (second and third heating run) of the sample was acquired from −30 °C to 220 °C after rapid cooling from 200 °C at the heating rates of 20 °C/min and 5 °C/min. All of the experiments were performed under a nitrogen atmosphere with the nitrogen flow rate of 50 mL/min, using aluminum sample pans. The glass transition temperature (*T_g_*) taken as the midpoint of the heat capacity change was obtained from the second calorimetric trace, the cold crystallization temperature (*T_cc_*), the melting temperature (*T_m_*) taken as the peak temperature maximum of the melting endotherm and the enthalpy values (*ΔH_cc_* and *ΔH_m_*) were obtained from the first calorimetric trace (first heating run) and for the amorphous samples from the second and third calorimetric traces (second heating runs with different heating rates) after rapid cooling from the melt (220 °C).

## 3. Results and Discussion

### 3.1. Properties of 3D-Printed Cosmetic Containers

Previous studies have shown that the processing conditions, in particular the contact time with the 3D printer platform, and smaller specimen’s surface contact area lead to an increase in crystalline phase during printing. During hydrolytic degradation, both crystallinity and weaker cohesion between the two printed layers of transverse pattern caused changes during degradation. The build direction and layer orientation proved to be a more important parameter conditioning degradation than the hydrophobicity of the specimens. In general, the direction of 3D printing is an important parameter that should be taken into account when designing 3D printed material applications [12,37], therefore the prototype of cosmetic containers have been carefully examined to determine the differences in the properties of their individual parts—the jar bottom or lid top as well as walls.

In order to evaluate the changes in the thermal properties of the tested samples as a consequence of the thermal history during the processing by 3D printing, DSC analysis was conducted (Table 2).

The DSC results of the PLA filament show only a glass transition temperature suggesting that the filament is amorphous. The presence of PHA component in the filament not only plasticizes but also initiate the crystallization of the blend by nucleation as the nucleation agent (Table 2). Likewise during 3D printing, tensile forces acted upon the filament, resulting in a stress-induced crystallization and orientation of the layers of printed elements, and also individual parts of the cosmetic container had a different contact time with the printer platform [2,12]. The jar bottom or lid top had a longer contact time (15–18 min) and were subjected to a higher temperature for longer time. The walls, away from the platform, were less influenced by temperature, which affected the thermal properties of these parts. The melting and cold crystallization enthalpies for the PLA jar bottom were higher (*ΔH_m_* = 2.45 J/g and *ΔH_cc_* = 2.92 J/g) than for the jar wall (*ΔH_m_* = 1.88 J/g and *ΔH_cc_* = 1.79 J/g), which indicates that the processing, especially contact time with printer platform causes an increase in the crystalline phase of these parts of the jar. For the PLA/PHA blend, these effects were slightly lower than in the case of PLA material. After 3D printing of cosmetic containers results show that the material crystallizes during heating. In the second heating run at 20 °C/min, crystallization and melting effects are low, which indicates slow nucleation and crystallization.

A slower first heating rate of 5 °C/min for the PLA/PHA blend (data not shown) allows nucleation and crystal growth in the sample to give a multiple melting endotherm with three *T_m_* values (143.8 °C, 152.5 °C and 173.4 °C). Further extension of the crystallization time causes only in a slight decrease in the cold crystallization temperature from 94.1 °C at 5 °C/min to 89.5 °C at 2.5 °C/min.

The processing conditions, in particular the printing orientation of individual parts of the container, influenced the material properties, which can then affect the time of use and degradation process of these individual parts, especially their disintegration.

### 3.2. Accelerated Aging Test

Long-term storage is simulated by an accelerated shelf life study. Accelerated aging is an artificial procedure that allows to determine the lifespan or shelf-life of the product at an accelerated pace [29]. According to the standard, it is “storage of samples at an elevated temperature in order to simulate real time aging in a reduced amount of time” [38]. The data obtained in the study are based on conditions simulating the effects of aging of materials. The results of the accelerated aging test of the packaging-product system simulate the requested period up to the expiry date of the product (3 months, 6 months, one year, etc.). Accelerated aging data is accepted only if those tests can be repeated in real time and demonstrates the stability of both packaging and product materials over time. This method is widely used in stability tests of pharmaceutical and food packaging [30]. The progress of material aging was estimated by material examination and failure analysis (macroscopic observations of the specimens’ surfaces), the specimens’ molar mass and thermal properties changes during the performed experiments.

A preliminary test of accelerated aging at 55 °C of prototype PLA packages using cosmetic simulants (paraffin, ethanol, deionized water and blank test) performed over a period of 37 days (real-time aging of one year) have shown that cosmetic containers were deformed from the beginning of experiment. In the case of paraffin and blank test, the concavity of the cosmetic containers was found, in the case of ethanol swelling, and in the case of deionized water—both the concavity and swelling. Deionized water and ethanol evaporated, however ethanol evaporated more slowly, because the swelling cosmetic container kept its tightness. Figure 1 shows photomacrographs of the containers after 19 days of degradation. After this time, the deformation did not change much.

Materials from PLA easily deformed at low temperatures due to the relatively low glass transition temperature between 55 °C and 65 °C [39]. The PHA component of the PLA/PHA blend leads to a significant reduction in deformation during aging [12]. A higher degree of crystallinity of the PLA/PHA cosmetic container may increase the stability of the material at elevated temperature [40]. For further research, only paraffin was chosen which does not evaporate so quickly.

The changes in the thermal properties after 37 days (real-time aging of one year) of preliminary accelerated aging test of PLA cosmetic containers and after 185 days (real-time aging of five years) of accelerated aging test of both PLA and PLA/PHA cosmetic containers were examined using DSC analysis (Table 3).

Melting enthalpies increase during aging of the PLA cosmetic containers filled with cosmetic simulants occurred in the following order: jar with ethanol > jar with paraffin > jar with water, indicating that the aging with cosmetic simulants causes an increase in the crystalline phase of the jars. The largest increase was noted for aging with ethanol, *ΔH_m_* = 33.07 J/g (before aging *ΔH_m_* = 2.45 J/g) together with a slight decrease in *T_g_*, which indicates the highest increase in the crystalline phase of this material and thus the highest degradation, which is also confirmed by the GPC analysis (see Figure 3). Also, for this material there is no cold crystallization effect. A cold crystallization exotherm can be detected at temperatures between *T_g_* and *T_m_* and commonly occurs in the DSC trace of amorphous to partially crystallized semicrystalline polymers when the polymers are heated to a temperature above *T_g_*, at which the crystallizable polymer chains possess enough segmental mobility to crystallize. The crystallinity of PLA and PLA/PHA samples after 185 days of degradation with paraffin, and blank test increased after a long time due to the degradation process. The cold crystallization enthalpy of samples aging after 37 days in the paraffin, water and blank test almost did not differ from the consecutive melting enthalpies, confirming that the PLA cosmetic containers were amorphous. The cold crystallization phenomenon was observed during the heating run at 5 °C/min for all amorphous samples received previously after rapid cooling from 200 °C.

During the first heating run, for PLA cosmetic containers aging with ethanol and paraffin were noted the multiple melting endotherms. The multiple melting endotherm in the DSC curves has been reported for many semicrystalline polymers crystallized isothermally from the melt at a selective crystallization temperature. There is no single explanation for this effect. In our case it may be a consequence of (i) melting, recrystallization, and remelting during the DSC heating process; (ii) changes in the morphology (such as lamellar thickness, distribution, crystal perfection or stability); (iii) physical aging or/and relaxation of the rigid amorphous fraction; (iv) different molar mass of individual polymer chains as a result of the degradation mechanism that can lead to the formation of a multiple population of crystallites, but only in the case of aging with ethanol, where a relatively high degree of degradation was observed (see Figure 3) [6,41]. In the case of PLA samples after 185 days of degradation with paraffin, and blank test, the double melting endotherm indicates most likely different molar mass of individual polymer chains resulting from the degradation mechanism. During the second heating run after a rapid cooling, for aging with deionized water, and for blank test for both aging times (37 and 185 days) the PLA samples had no chance of nucleation during cooling and thus only showed *T_g_*. Samples analyzed at 20 °C/min do not crystallize likely due to low degradation degree. Heating rate of 5 °C/min allows the crystallization of all materials. During the second heating run at 20 °C/min, for aging PLA with ethanol after 37 days and with paraffin after 185 days as well as for aging of PLA/PHA samples after 185 days, except for the blank test, the crystallization and melting effects are low, which indicates slow nucleation and crystallization.

The molar mass loss of cosmetic containers during aging using cosmetic simulant paraffin and blank test occurred in the following order: PLA with paraffin > PLA/PHA blank test > PLA blank test > PLA/PHA with paraffin, respectively (Figure 2).

We originally demonstrated, that degradation of PLA film samples occurs in paraffin due a residual water content [3,6]. The same effect causes a faster degradation of the PLA cosmetic containers filled with paraffin (Figure 3). The residual content of water can be absorbed from hydrophobic paraffin and the environment by the dry PLA container and penetrate into polymer matrix, generating an autocatalytic effect. Faster internal degradation of polylactide is regarded as a general phenomenon [3,42]. Paraffin also has a heat buffering capacity and it has also been proven that it has a cooling effect [43,44], which is why it slowed the degradation of the temperature-sensitive PHA in the blend and thus the degradation of entire container.

Hydrolysis and alcoholysis may share the same mechanism, but differ in acyl acceptors (water or ethanol) and may occur without catalysts. The highest degree of degradation occurred in the case of ethanol. For deionized water, this degree was small, due to the largest deformation of the jar and evaporation of water during degradation. Performed studies for both materials—PLA and PLA/PHA blend—have shown that degradation of those material accelerates after 111 days of aging with paraffin, and for blank test (real-time aging of three years), which is advantageous from the viewpoint of products with a long shelf-life application of three years.

### 3.3. 3D-Printed Container/Cosmetic Formulation Compatibility Test

The compatibility tests were based on determining the interaction of the cosmetic formulation with the packaging in the surrounding environment in order to verify their fit. The packaging should be compatible with the product, which means that all ingredients of the cosmetic formulation do not affect the packaging and vice versa, the packaging components do not react with the cosmetic formulation. The basis for the selection of the correct packaging is knowledge about the ingredients of the recipe. The evaluation criteria taken into account for both packaging and cosmetic formulation are appearance, color and odor [45].

After 12 weeks of compatibility test for PLA cosmetic containers, similar relationships were observed for all samples. The jars were highly swollen, and the cosmetic formulations leaked or evaporated, so that the tests were negative. In the case of PLA/PHA cosmetic containers in the middle of the compatibility test (after 6 weeks) the first changes, such as a slight opening of the jars and a small mass loss (vaporization of 10% for ambient temperature and from 15% to 20% for 45 °C) only for more moisturizing cosmetic formulations with no changes in color or odor as well as with no deformations or swelling of the jars could be noted (Table 4). After 12 weeks at ambient temperature, regardless of the type of cosmetic formulation, the jars have opened more with a mass loss of 50–70% for more moisturizing and 30–40% for oily cosmetic formulations. For the cosmetic containers with cosmetic formulations incubated at 45 °C, a significant mass loss of 75–80% for more moisturizing and 60–75% for oily cosmetic formulations were noticed with a slight change in color (darkening) as well as an odor (unpleasant, pungent) in one oily formulation and significant vaporization for one moisturizing formulation (Table 4). Odor change did not occur during tests in conventional polypropylene cosmetic containers, although the change in color to darker depends mainly on temperature (own research by Dr. Irena Eris Cosmetic Laboratories). During the compatibility test, the deformation of cosmetic containers was in the range of 1–5%. No cracks occurred, and the color of the jars did not change.

The packaging/cosmetic formulation compatibility test carried out was negative for PLA and positive for PLA/PHA cosmetic containers, which proved that the blend with the addition of PHA component has better compatibility with real cosmetic formulations.

### 3.4. Cosmetic Contamination Simulation—Effect on the Course of (Bio)degradation

In order to investigate the impact of cosmetic contamination on the course of degradation under laboratory composting conditions and under natural weathering conditions, a comparative degradation test of empty cosmetic containers (blank test) and cosmetic containers with a small amount of paraffin were carried out.

Ultimate biodegradation is the decomposition of organic compounds caused by microorganisms, in aerobic conditions to CO_2_, water and mineral salts (mineralization) and new biomass, while in anaerobic conditions to CO_2_, methane (CH_4_), mineral salts and new biomass. In order to confirm the biodegradability of packaging materials and their ability to disintegrate under the influence of enzymes produced by bacteria and fungi, preliminary biodegradation tests were carried out under simulated composting conditions, in accordance with the method specified in PN-EN 14806: 2010 [46] and US ASTM standard D 6400 [35]. This method simulates ambient conditions found in industrial composting plants. The biodegradation tests were performed under laboratory conditions simulating aerobic composting. The CO_2_ released during biodegradation of the samples was measured in accordance with the test procedure at constant process parameters. The calculated degree of biodegradation and mass loss of PLA and PLA/PHA cosmetic containers during aerobic laboratory composting are presented in Table 5.

The degree of biodegradation of cosmetic containers during laboratory composting occurred in the following order: PLA jar with paraffin > PLA/PHA jar with paraffin > PLA/PHA empty jar > PLA empty jar, respectively. The degradation of polyesters during organic recycling in biological environments, including anaerobic and aerobic conditions, results from enzymatic attack or simple hydrolysis or both [47]. PLA is widely considered as a biodegradable polymer, however, poly(*α*-hydroxy acid)-type polyesters are being proven to degrade in the compost rather as a result of simple chemical hydrolysis occurring relatively quickly in these conditions, despite the fact that some fungi strains, such as *Penicillium verrucosum* and *Aspergillus ustus* can cleave their main chain [7,48]. Therefore, the degradation of empty PLA cosmetic containers during laboratory composting was slower than for those with the addition of PHA. The presence of microbial-derived PHA accelerates the degradation process. The degree of biodegradation of PLA and PLA/PHA cosmetic containers during laboratory composting after 84 days was higher for cosmetic containers with paraffin simulating cosmetic contamination.

The changes in the thermal properties after 84 days of laboratory composting and after 128 and 365 days of the natural weathering conditions of PLA and PLA/PHA cosmetic containers were examined using DSC analysis (Table 6).

The melting enthalpies increase together with a decrease in *T_g_* and *T_m_* during laboratory composting of the cosmetic containers especially for samples filled with paraffin, indicating that the paraffin induce (bio)degradation and an increase in the crystalline phase of the jars. Only PLA cosmetic container filled with paraffin exhibited a cold crystallization effect during the first heating run at 20 °C/min. It may be a plasticization effect (by degradation products) that reduces *T_g_* and *T_m_* and allows crystallization in the first heating run. Other samples analyzed at 20 °C/min do not crystallize. Heating rate of 5 °C/min allows the crystallization of all materials. During the second heating run at 20 °C/min, all the examined samples (amorphous samples previously obtained by rapid cooling from 200 °C) exhibited a cold crystallization effect. However, for PLA cosmetic container (bio)degradation, crystallization and melting effects are low, which indicates slow nucleation and crystallization and therefore slow (bio)degradation. It is well known that the *T_g_* of a polymer depends on the length of the chain. It can be observed that, for the PLA cosmetic container filled with paraffin a significant decrease in *T_g_*, of approximately 52%, accompanied a further decrease in mass-average molar mass *M_w_* up to 7000 g/mol (96% of molar mass loss, see Figure 4). The same effect was noticed for other samples with comparable amount of biodegradation degree: PLA/PHA empty jar and PLA/PHA jar filled with paraffin. In the first heating run for a PLA cosmetic container filled with paraffin a multiple melting endotherm was noted. This effect did not occur in the second heating run due to different sample thermal history.

In contrast to the degradation under laboratory composting conditions, samples degraded under natural weathering conditions have not been noted any significant changes in the thermal properties. Only in the case of cosmetic containers from the PLA/PHA blend, a slight increase in melting enthalpies value was observed along with a slight decrease in *T_g_* and *T_m_*. During the second heating run after a rapid cooling, for cosmetic containers from the PLA and PLA filled with paraffin the samples had no chance of nucleation during cooling and thus only showed *T_g_*. Samples analyzed at 20 °C/min do not crystallize. Heating rate of 5 °C/min allows the crystallization of those samples, just like for the other materials. For cosmetic containers from the PLA/PHA blend, with and without paraffin in the second heating run, crystallization and melting are low, which indicates slow nucleation, and crystallization.

Both, under composting conditions (laboratory compost, composting pile and BIODEGMA system), and under natural weathering conditions, the effect of paraffin contamination on the rate of degradation was observed (Table 6, Figure 4 and Figure 5). Under the influence of an environment rich in enzymes produced by bacteria and fungi, paraffin accelerated the (bio)degradation as an additional carbon nutrient and energy source, while where it was the influence of natural weathering condition, in which temperature and humidity had a major role, slowed down. The molar mass loss of cosmetic containers during laboratory composting occurred in the following order: PLA jar with paraffin contamination > PLA/PHA jar with paraffin contamination > PLA/PHA empty jar (blank test) > PLA empty jar (blank test), respectively (Figure 4A), what correlated with biodegradation degree ordering. While the order under natural weathering conditions was PLA/PHA empty jar (blank test) > PLA empty jar (blank test) > jars with paraffin contamination, however, these changes are diminutive (data not shown).

The degradation progress, comparing the composting environments, after 21 days for both cosmetic containers (with paraffin and empty) was faster in the BIODEGMA system than in the composting pile (data not shown). Whereas, after 84 days the degradation of cosmetic containers with paraffin contamination was at the same level independently from environment: composting pile or laboratory compost. Then, for empty containers, the degradation for PLA/PHA occurred in the order: laboratory compost and composting pile (Figure 4B) and vice versa for PLA, because more humid environments promotes faster hydrolytic degradation of PLA material. In general, degradation in the humid compost environments was in favor of PLA containers (data not shown).

Macroscopic visual evaluation of the PLA and PLA/PHA cosmetic containers after degradation under laboratory and industrial composting conditions showed erosion through the breaking of the specimens, especially at the point of contact of the wall with the bottom of the jar, where cohesion between two adjacent printed layers was weaker. However, this effect was more visible for containers incubated in industrial composting pile (Figure 5). Parts with the largest structural ordering of the material as the jar bottom and lid degrade slower compare with jar wall. For cosmetic containers with the addition of paraffin disintegration occurs much faster because paraffin contamination accelerates the degradation. Material containing PHA is usually less transparent (yellowish) than PLA itself, however, during degradation of the PLA-based material, a decrease in transparency and milky-white color was observed due to increased crystallinity [3,12].

Both the ex-ante investigations as well as the ex-post studies are needed in the area of advanced polymer material applications (especially of long-shelf life products such as cosmetics or household chemicals) in order to increase efficiency and to define and minimize the potential failure of novel (bio)degradable polymer products before and after specific use. Comprehensive predictive studies may help to design novel polymeric materials and to avoid the failures of existing ones. The modeling and simulations of degradation using the tests performed allow predicting the behavior of advanced materials during their use and disposal. Knowledge of degradation and damage phenomena of (bio)degradable polymer materials in service conditions and thus the prediction of behavior under operating conditions, indicates the scope and capabilities as well as limitations of using these polymers as advanced polymeric materials. The evaluation protocol summarizing the work is presented in Table 7.

## 4. Conclusions

Based on the knowledge of the cosmetic formulation and packaging material, as well as on the basis of available standards for the packaging of medical, pharmaceutical, and food products, a comprehensive strategy of standard procedure for cosmetic products was developed to characterize the degradation pathways of polymeric systems (PLA and PLA/PHA containers made with 3D printing). These procedures involved virtual tests under different environmental conditions, including accelerated aging test, packaging/cosmetic formulation compatibility test with real cosmetic formulations, composting under laboratory, and industrial conditions as well as weathering test. The performed study of modeling and simulation of degradation and damage phenomena have shown that: (i) microbial-based PHA component during accelerated aging test at 55 °C and during incubation in environments rich in microorganisms accelerated the degradation process of material investigated; (ii) paraffin accelerates PLA degradation due to the residual water content and slows down PHA-based material degradation because of paraffin heat buffering capacity and a cooling effect; (iii) under the influence of an environment rich in enzymes produced by bacteria and fungi, paraffin contamination accelerates the biodegradation; (iv) under the influence of natural weathering condition paraffin contamination slowed down degradation due to its hydrophobicity and water-repellent activity; (v) aging PLA and PLA/PHA container with paraffin, accelerated after 111 days corresponding to real time—three years, which is advantageous from the viewpoint of products with a long shelf-life applications. The processing conditions, in particular the printing orientation of individual parts of the container, influenced the material properties at its various regions, affecting the rate of degradation of individual parts and the simulated shelf life. Accelerated aging tests of prototype packages using cosmetic simulants have shown that cosmetic containers have been deformed from the beginning of experiments especially in the case of PLA material, so for PLA cosmetic containers the tests were negative. For PLA/PHA cosmetic containers, it has been proved that the material (in this case a blend) with the addition of more hydrophobic PHA component demonstrated better compatibility with the cosmetic formulations. The selection of test procedures allow to indicate limitations both in the preparation of the finished product package and its shelf life.

## Figures and Tables

**Figure 1 materials-12-00994-f001:**
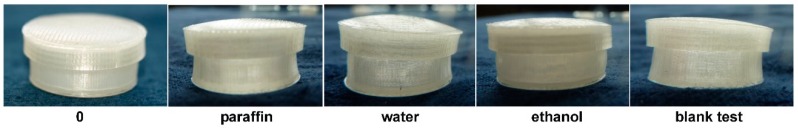
Photomacrographs of PLA cosmetic containers filled with paraffin, ethanol, deionized water, and blank test after 19 days of the accelerated aging test at 55 °C.

**Figure 2 materials-12-00994-f002:**
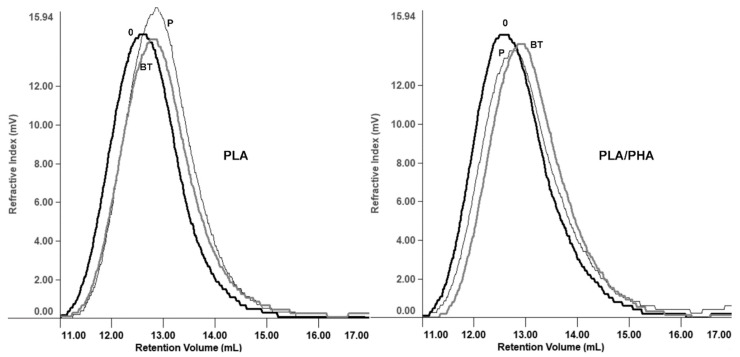
Overlay of selected GPC elugrams of PLA and PLA/PHA cosmetic containers filled with paraffin (P), and blank test (BT) before aging (0) and after 185 days (real-time aging of five years) of the accelerated aging test at 55 °C.

**Figure 3 materials-12-00994-f003:**
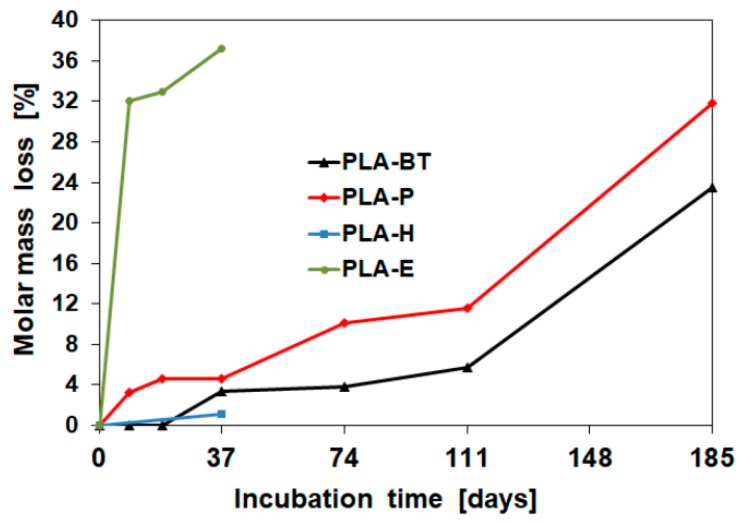
Molar mass loss of PLA cosmetic containers filled with paraffin (P), ethanol (E), deionized water (H), and blank test (BT) as a function of incubation time of the accelerated aging test at 55 °C. The molar mass loss is given as a percentage of the original mass-average molar mass.

**Figure 4 materials-12-00994-f004:**
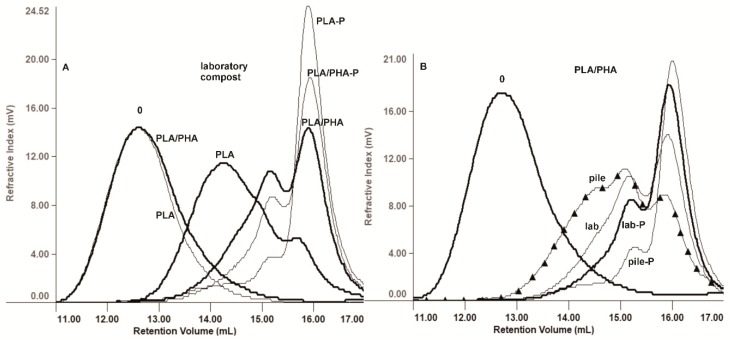
Overlay of selected GPC elugrams of (**A**): PLA and PLA/PHA cosmetic containers before degradation (0) and after 84 days of degradation in the laboratory compost and (**B**): PLA/PHA cosmetic containers before degradation (0) and after 84 days of degradation in the laboratory compost (lab) and in the composting pile (pile); P—container with paraffin contamination.

**Figure 5 materials-12-00994-f005:**
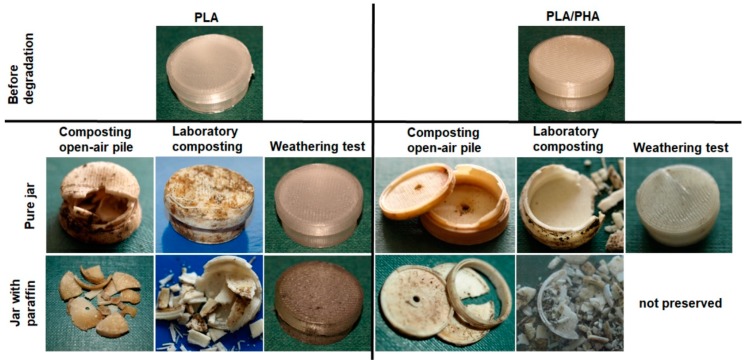
Photomacrographs of PLA cosmetic containers before degradation and after 84 days of the composting under laboratory and industrial conditions as well as after 365 days under natural weathering conditions.

**Table 1 materials-12-00994-t001:** Printer settings for cosmetic containers fabrication.

Printer Settings	
Nozzle diameter (mm)	0.4
Layer height (mm)	0.1
Infill pattern	rectilinear
Infill density (%)	100
Print speed (mm/s)	50
Head travel speed (mm/s)	100
Nozzle temperature (°C)	195
Platform temperature (°C)	65
Printing time (min)	15–18

**Table 2 materials-12-00994-t002:** Selected calorimetric parameters of PLA and PLA/PHA filament and cosmetic container parts. I-heating run and II-heating run (after rapid cooling from 200 °C at 20 °C/min).

Variable	PLA Filament	PLA Jar Wall	PLA Jar Bottom	PLA/PHA Filament	PLA/PHA Jar Wall	PLA/PHA Jar Bottom
I-heating run						
*T_m_* (°C)	-	152.0	151.2	152.2/172.8	154.0/172.0	151.8/171.1
*ΔH_m_* (J/g)	-	1.88	2.45	18.50	20.94	21.0
*T_cc_* (°C)	-	133.2	129.3	111.9	116.2	114.1
*ΔH_cc_* (J/g)	-	−1.79	−2.92	−18.36	−20.63	−20.55
II-heating run						
*T_g_* (°C)	61.8	61.9	61.1	1.7/60.5	1.7/59.6	2.0/59.7
*Δcp* (J/g°C)	0.50	0.47	0.52	0.05/0.49	0.06/0.46	0.06/0.46
*T_m_* (°C)	-	153.3	149.6	154.4/175.1	153/173.3	154.1/173.6
*ΔH_m_* (J/g)	-	0.06	0.38	0.40	0.70	0.65
*T_cc_* (°C)	-	140.5	138.2	134.4	142.1	142.5
*ΔH_cc_* (J/g)	-	−0.08	−0.32	−0.36	−0.68	−0.62

*T_g_*—glass transition temperature, *Δcp*—the increment of heat capacity at the glass transition, *T_m_*—melting temperature, *ΔH_m_*—melting enthalpy, *T_cc_*—maximum of the exothermic peak of the cold crystallization temperature, *ΔH_cc_*—cold crystallization enthalpy.

**Table 3 materials-12-00994-t003:** Calorimetric parameters of PLA cosmetic containers filled with paraffin (P) ethanol (E), deionized water (H), and blank test (BT) before degradation (jar bottom), and after 37 days of the accelerated aging test as well as PLA/PHA cosmetic containers filled with paraffin (P), and blank test (BT) before degradation (jar bottom), and after 185 days of the accelerated aging. First calorimetric trace was acquired at 20 °C as well as second and third calorimetric traces (after rapid cooling from 200 °C) at 20 °C/min and 5 °C/min, respectively).

Variable	PLA Jar Bottom	PLA-37P	PLA-37E	PLA-37H	PLA-37BT	PLA-185P	PLA-185BT	PLA/PHA Jar Bottom	PLA/PHA-185P	PLA/PHA-185BT
I-calorimetric trace, 20 °C/min										
*T_m_* (°C)	151.2	155.2/169.2/175.4	142.2/149.1/152.5/157.5	152.4	151.5	127.9/152.1	127.3/152.8	151.8/171.1	153.8/170.0	154.3/170.2
*ΔH_m_* (J/g)	2.45	5.47	33.07	3.87	2.90	1.47/23.22	1.74/23.77	21.0	28.9	30.1
*T_cc_* (°C)	129.3	116.4	-	127.8	121.2	-	-	114.1	-	-
*ΔH_cc_* (J/g)	−2.92	−5.03	-	−3.76	−2.95	-	-	−20.55	-	-
II-calorimetric trace after rapid cooling, 20 °C/min										
*T_g_* (°C)	61.1	61.5	58.0	62.0	61.9	60.4	62.2	2.0/59.7	1.3/57.7	1.0/57.0
*Δcp* (J/g°C)	0.52	0.49	0.50	0.49	0.49	0.52	0.48	0.06/0.46	0.04/0.47	0.04/0.47
*T_m_* (°C)	149.6	149.4	152.1	-	-	150.9	-	154.1/173.6	153.3/170.3	154.1/170.1
*ΔH_m_* (J/g)	0.38	0.19	0.44	-	-	0.37	-	0.65	0.79	4.19
*T_cc_* (°C)	138.2	-	139.1	-	-	139.1	-	142.5	139.0	135.1
*ΔH_cc_* (J/g)	−0.32	-	−0.42	-	-	−0.34	-	−0.62	−1.18	−4.11
III-calorimetric trace after rapid cooling, 5 °C/min										
*T_g_* (°C)	58.7	58.4	55.7	58.0	58.2	57.6	58.1	−0.9/56.1	−0.2/54.9	−0.1/54.8
*Δcp* (J/g°C)	0.56	0.49	0.54	0.47	0.48	0.54	0.5	0.05/0.50/	0.04/0.48	0.04/0.47
*T_m_* (°C)	150.0	150.0	148.8/153.1	151.0	150.8	150.7	151.6	150.4/172.1	149.8/154.2/171.5	150.0/154.4/172.0
*ΔH_m_* (J/g)	10.21	6.26	26.28	9.63	7.51	18.25	12.21	20.50/5.25	22.76/5.47	33.64
*T_cc_* (°C)	123.5	125.8	118.1	125.9	126.2	122.3	126.2	118.7	116.4	113.5
*ΔH_cc_* (J/g)	−10.21	−6.20	−25.93	−9.40	−7.50	−18.21	−12.03	−24.81	−28.14	−33.33

*T_g_*—glass transition temperature; *Δcp*—the increment of heat capacity at the glass transition; *T_m_*—melting temperature; *ΔH_m_*—melting enthalpy; *T_cc_*—maximum of the exothermic peak of the cold crystallization temperature; *ΔH_cc_*—cold crystallization enthalpy.

**Table 4 materials-12-00994-t004:** Compatibility test of PLA/PHA cosmetic containers with cosmetic formulations from the moisturizing to oily performed at ambient temperature and at 45 °C during 6 and 12 weeks.

Cosmetic Formulation	Viscosity (mPa·s)	Density (g/mL)	pH	Mass Loss (%)				Appearance Changes		Odor Changes	
				6 weeks		12 weeks		6 weeks	12 weeks	6 weeks	12 weeks
				AT	45 °C	AT	45 °C	45 °C		45 °C	
1	2300–3000	0.98–1.01	5.5–6.5	10	20	70	80	NC	CC	NC	MC
2	2400–3500	0.98–1.02	5.0–6.0	10	15	50	75	NC	CC	NC	MC
3	2400–4400	0.97–0.99	6.0–7.0	10	20	60	80	NC	CC	NC	MC
4	3000–5000	0.98–1.00	6.0–6.9	NC	NC	40	60	NC	CC	NC	SC
5	3800–4500	0.98–1.10	5.5–6.5	NC	NC	30	75	NC	CC	NC	MC

AT—ambient temperature; NC—no changes; CC—colour change; MC—minor change; SC—substantial change.

**Table 5 materials-12-00994-t005:** Biodegradation results after 84 days of laboratory composting.

Sample	The Degree of Biodegradation (%)	Mass Loss (%)
PLA empty jar (blank test)	21	5
PLA jar with paraffin (cosmetic contamination simulation)	34	10
PLA/PHA empty jar (blank test)	30	6
PLA/PHA jar with paraffin (cosmetic contamination simulation)	32	10

**Table 6 materials-12-00994-t006:** Calorimetric parameters of PLA and PLA/PHA cosmetic containers filled with paraffin (P), and blank test (BT) before degradation (jar bottom), after 84 days of the laboratory composting and after 128 and 365 days of the natural weathering conditions. First calorimetric trace was acquired at 20 °C as well as second and third calorimetric traces (after rapid cooling from 200 °C) at 20 °C/min and 5 °C/min, respectively).

Variable	PLA Jar Bottom	PLA-84	PLA-84P	PLA-365	PLA-365P	PLA/PHA Jar Bottom	PLA/PHA-84	PLA/PHA-84P	PLA/PHA-182	PLA/PHA-182P	PLA/PHA-365
I-calorimetric trace, 20 °C/min											
*T_m_* (°C)	151.2	148.9	126.0/139.7	152.2	153.9/165.4	151.8/171.1	142.5/149.0	133.7/144.8	155.1/174.2	151.5/170.4	151.6/169.5
*ΔH_m_* (J/g)	2.45	29.55	32.31	1.14	3.68	21.0	45.20	51.73	21.50	23.61	24.25
*T_cc_* (°C)	129.3	-	83.0	131.9	119.8	114.1	-	-	112.4	108.6	102.7
*ΔH_cc_* (J/g)	−2.92	-	−4.73	−0.92	−3.65	−20.55	-	-	−21.32	−23.39	−24.15
II-calorimetric trace after rapid cooling, 20 °C/min											
*T_g_* (°C)	61.1	51.2	32.6	62.1	61.7	2.0/59.7	37.5	38.9	1.7/59.5	0.2/57.8	0.6/59.6
*Δcp* (J/g°C)	0.52	0.54	0.55	0.48	0.50	0.06/0.46	0.57	0.59	0.04/59.54	0.05/0.49	0.05/0.45
*T_m_* (°C)	149.6	145.3	127.5	-	-	154.1/173.6	130.8/137.2	132.0/148.9	153.0/171.3	151.8/171.5	153.7/172.2
*ΔH_m_* (J/g)	0.38	1.08	10.11	-	-	0.65	31.93	20.52	0.76	0.74	0.74
*T_cc_* (°C)	138.2	127.9	105.2	-	-	142.5	101.2	102.5	142.2	143.0	143.3
*ΔH_cc_* (J/g)	−0.32	−0.86	−9.94	-	-	−0.62	−31.80	−20.38	−0.59	−0.65	−0.66
III-calorimetric trace after rapid cooling, 5 °C/min											
*T_g_* (°C)	58.7	54.3	43.4	58.3	58.1	−0.9/56.1	35.3	37.8	−0.5/56.1	−0.8/55.9	−0.8/56.2
*Δcp* (J/g°C)	0.56	0.43	0.59	0.46	0.48	0.05/0.50/	0.57	0.58	0.05/0.43	0.05/0.48	0.04/0.46
*T_m_* (°C)	150.0	144.2/149.7	127.3/137.9	150.9	150.6	150.4/172.1	127.1/137.5/150.0	120.6/130.9/148.9	150.4/172.1	149.5/171.7	150.8/172.5
*ΔH_m_* (J/g)	10.21	28.55	40.49	4.87	6.60	20.50/5.25	39.06	34.82	24.43	25.61	24.74
*T_cc_* (°C)	123.5	115.3	86.0/95.8	126.3	125.6	118.7	87.4	89.3/96.0	118.8	118.6	119.7
*ΔH_cc_* (J/g)	−10.21	−28.50	−40.32	−4.80	−5.87	−24.81	−38.93	−34.70	−24.35	−24.63	−22.60

*T_g_*—glass transition temperature, *Δcp*—the increment of heat capacity at the glass transition, *T_m_*—melting temperature, *ΔH_m_*—melting enthalpy, *T_cc_*—maximum of the exothermic peak of the cold crystallization temperature, *ΔH_cc_*—cold crystallization enthalpy.

**Table 7 materials-12-00994-t007:** Testing of long-term (bio)degradable polymers.

Evaluation Protocol	
Material (filament) composition and characterization	GPC, NMR, TGA, DSC, FTIR, ESI-MS [12]
Sample design	CAD model
	Determining printing parameters/printer settings
	Fabrication of 3D-printed prototype items (cosmetic containers)
	Characterization of research material in the form of shaped articles (GPC, DSC)
	Determining effect of printing directions and layer orientation on the properties of 3D-printed specimens/individual parts (GPC, DMA, TGA, DSC, FTIR, SEM, optical microscope, tensile tests) [12,37]
Modeling and simulation of degradation and damage phenomena	Planning of experiments/selection of virtual testing processes
Investigation of degradation processes:	Time
Accelerated aging test	9, 19, 37, 74, 111 and 185 days corresponding to real-time aging 3, 6, 12, 24, 36 and 60 months respectively
Packaging/cosmetic formulation compatibility test	6 and 12 weeks
Composting under laboratory conditions/disintegration biodegradability test	12 weeks (84 days, according to USA standard ASTM D 6400)
Weathering test	12 months
Degradation under industrial composting conditions:	
BIODEGMA system	3 weeks
Static composting open-air pile	3 and 12 weeks
Characterization of the materials after degradation and data analyze/relationship between the structure, properties and degradation of long-term (bio)degradable polymers	

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
