# Peer review of "3D-Printed Polyester-Based Prototypes for Cosmetic Applications—Future Directions at the Forensic Engineering of Advanced Polymeric Materials"

_materials, 2019, doi:10.3390/ma12060994_

Reviewer 1 Report

Dear Authors,

congratulation to this paper.

I had only one recommendation:

In Table 7 you show to different sources:

Polym. Degrad. Stab.
152, 191-207 (2018); Polym. Degrad. Stab. 156,
100-110 (2018)

Please add here also the reference number.

BR

One reviewer

Author Response

Response to Reviewer 1 Comments

We are indebted for such a positive opinion of the Reviewer regarding our manuscript.

Point 1: In Table 7 you show to different sources: Polym. Degrad. Stab. 152, 191-207 (2018); Polym. Degrad. Stab. 156, 100-110 (2018). Please add here also the reference number.

Response 1: According to the recommendation regarding Table 7, the sources have been replaced with a reference numer.

Reviewer 2 Report

The manuscript entitled “3D printed polyester-based prototypes for cosmetic applications-future directions at the forensic engineering of advanced polymeric materials” by Rydz et al. investigates the preparation by 3D printing and the degradation of cosmetic packaging from biodegradable polymers as a function of storage conditions.

The paper is well written, clear and the conclusions are supported by the results. However, some corrections will improve the overall quality of the paper.

1.      Lines 51 and 60: the full name for PLA and PHA must be provided when these abbreviations are used for the first time in the text.

2.      Line 59: “…as modifiers of PLA…”

3.      Line 70: references must be added at the end of the sentence:”so far, …..structural components.”

4.      Line 94: references must be added at the end of the sentence:”in general,…or landfill disposal.”

5.      Line 105:”…the aim of this study…”

6.      Line 174: full name for “TLC”

7.      Lines 283-285: the sentence “a slight increase in Tg of both materials were also observed (table 2) as an result of the increase in crystalline domains” is not very clear and correct. Which are the two materials and what are the Tg values?! The Tg values given in table 2 are almost constant for both PLA and PLA/PHA materials.

8.      Figure 1 and lines 306-312: the results given in this figure and this paragraph where performed for 37 days (given in text) or 19 days (given in the caption of fig 1)?!

9.      Line 315: what are the values of “low temperatures”?!

10.  Line 339 and 343-344: a repetition is made in these two sentences: “also for this material there is no cold crystallization effect” and “the cold crystallization …was not observed.”

11.  Lines 345-346: the authors should clarify if the crystallization increase is a consequence of the degradation or the degradation is a consequence of the crystallization increase?!

12.  Lines 333-350: in this paragraph the authors should discuss the obtained results as a function of the initial PLA and PLA/PHA filaments calorimetric parameters.

13.  Line 352: “the multiple melting endotherms…”

14.  Line 365:”samples analyzed at 20˚C…”

15.  Line 396: “the compatibility tests are/were based on…”

16.  Line 455: delete “of laboratory composting”

17.  The correlations between the data provided in table 6 are very difficult to follow. I think that a figure illustrating the variation of Tm as a function of time would be more appropriate.

18.  Line 473: “other samples analyzed at…”

19.  Line 476: delete “biodegradation of” and put “,” after “container”

20.  Line 482: “…with comparable amount of biodegradation degree.”

21.  Line 484-485: can the authors explain why the multiple melting endotherms do not longer appeared after the second heating run?!

22.  Line 491: “samples analyzed at…”

23.  Lines 491-492: as these effects was already observed under the laboratory composting conditions, I think that the authors must state this fact by adding something like that :”similar to the results obtained after laboratory composting conditions…”

24.  Line 504: I propose: “while the order under natural weathering conditions was:…”

25.  Line 554: replace “degradation” with “analysis” or “storage”

26.  Line 55: delete “in paraffin”

27.  Line 559: delete the second “paraffin”

28.  Line 561: delete “of aging”

In view of the above I recommend the publication of the manuscript in Materials after major corrections.

Author Response

Response to Reviewer 2 Comments

We are indebted for the opinion of the Reviewer regarding our manuscript. Answering the specific comments we would like to state that:

Point 1:      Lines 51 and 60: the full name for PLA and PHA must be provided when these abbreviations are used for the first time in the text.

Response 1: According to the suggestion of the Reviewer, the abbreviations have been provided.

Point 2:      Line 59: “…as modifiers of PLA…”

Response 2: According to the suggestion of the Reviewer, sentence have been corrected.

Point 3:      Line 70: references must be added at the end of the sentence:”so far, …..structural components.”

Response 3: According to the suggestion of the Reviewer, references have been added at the end of the sentence.

Point 4:      Line 94: references must be added at the end of the sentence:”in general,…or landfill disposal.”

Response 4: According to the suggestion of the Reviewer, references have been added at the end of the sentence.

Point 5:      Line 105:”…the aim of this study…”

Response 5: According to the suggestion of the Reviewer, sentence have been corrected.

Point 6:      Line 174: full name for “TLC”

Response 6: According to the suggestion of the Reviewer, sentence have been corrected.

Point 7:      Lines 283-285: the sentence “a slight increase in Tg of both materials were also observed (table 2) as an result of the increase in crystalline domains” is not very clear and correct. Which are the two materials and what are the Tg values?! The Tg values given in table 2 are almost constant for both PLA and PLA/PHA materials.

Response 7: In fact, some mistake had to get into this sentence so it was removed.

Point 8:      Figure 1 and lines 306-312: the results given in this figure and this paragraph where performed for 37 days (given in text) or 19 days (given in the caption of fig 1)?!

Response 8: A preliminary test of accelerated aging at 55 °C of prototype PLA packages have been performed over a period of 37 days, but containers were deformed from the beginning of the experiment. For clarity, the following sentences have been added to the text:

Figure 1 shows photomacrographs of the containers after 19 days of degradation. After this time, the deformation did not change much.

Point 9:      Line 315: what are the values of “low temperatures”?!

Response 9: “Plastics consist of long chain molecules that are entangled with one another. The degree of entanglement varies with the length and exact shape of the polymer molecule. Long chains in most polymers at moderate temperatures are able to slither over one another and the material is flexible and does not crack – they are considered to be in a ‘rubbery’ state. As the temperature is decreased, most polymers begin to stiffen up and they go through what is known as the ‘glass transition’ to become effectively glassy solids with all the properties of glass i.e. they become very hard and also very brittle.” Thus by “low temperatures for polymers” it is consider any temperature below Tg for the particular polymer.

Point 10:  Line 339 and 343-344: a repetition is made in these two sentences: “also for this material there is no cold crystallization effect” and “the cold crystallization …was not observed.”

Response 10: According to the suggestion of the Reviewer, repetition have been deleted.

Point 11:  Lines 345-346: the authors should clarify if the crystallization increase is a consequence of the degradation or the degradation is a consequence of the crystallization increase?!

Response 11: the crystallinity of PLA and PLA/PHA samples after 185 days of degradation with paraffin, and blank test increase after a long time due to the degradation process.

According to the suggestion of the Reviewer, sentence have been clarified.

Point 12:  Lines 333-350: in this paragraph the authors should discuss the obtained results as a function of the initial PLA and PLA/PHA filaments calorimetric parameters.

Response 12: PLA and PLA/PHA filaments were not degraded. The initial material under the study was a prototype container (PLA and PLA/PHA jar) before degradation and that is why we referred to it. Detailed studies regarding the filament can be found in the publication: Gonzalez Ausejo, J.; Rydz, J., et al. A comparative study of three-dimensional printing directions: The degradation and toxicological profile of a PLA/PHA blend. Polym. Degrad. Stab. 2018, 152, 191–207 (reference 12).

Point 13:  Line 352: “the multiple melting endotherms…”

Response 13: According to the suggestion of the Reviewer, sentence have been corrected.

Point 14:  Line 365:”samples analyzed at 20˚C…”

Response 14: According to the suggestion of the Reviewer, sentence have been corrected.

Point 15:  Line 396: “the compatibility tests are/were based on…”

Response 15: According to the suggestion of the Reviewer, sentence have been corrected.

Point 16:  Line 455: delete “of laboratory composting”

Response 16: According to the suggestion of the Reviewer, sentence have been corrected.

Point 17:  The correlations between the data provided in table 6 are very difficult to follow. I think that a figure illustrating the variation of Tm as a function of time would be more appropriate.

Response 17: Due to the fact that some Tm values are double (multiple melting endotherms), presentation of the figure illustrating the variation of Tm as a function of time is impossible.

Point 18:  Line 473: “other samples analyzed at…”

Response 18: According to the suggestion of the Reviewer, sentence have been corrected.

Point 19:  Line 476: delete “biodegradation of” and put “,” after “container”

Response 19: According to the suggestion of the Reviewer, sentence have been corrected.

Point 20:  Line 482: “…with comparable amount of biodegradation degree.”

Response 20: According to the suggestion of the Reviewer, sentence have been corrected.

Point 21:  Line 484-485: can the authors explain why the multiple melting endotherms do not longer appeared after the second heating run?!

Response 21: The multiple melting endotherms depends on sample thermal history. In the second heating run the thermal history of the sample were quite different then in the first heating run (were the sample was crystalline). The sample after melting in first heating run were rapid cooled to obtain the amorphous sample. Next in the second heating run the amorphous sample (not crystalline) were heating to determine the glass transition temperature. Above the glass transition temperature during further heating the cold crystallization of the sample and then the melting were observed.

The sentence has been completed to clarify it.

Point 22:  Line 491: “samples analyzed at…”

Response 22: According to the suggestion of the Reviewer, sentence have been corrected.

Point 23:  Lines 491-492: as these effects was already observed under the laboratory composting conditions, I think that the authors must state this fact by adding something like that :”similar to the results obtained after laboratory composting conditions…”

Sentence: “Heating rate of 5 °C/min allows the crystallization of all materials.” is connected to sentence:

Response 23: According to the suggestion of the Reviewer, sentence have been corrected.

Point 24:  Line 504: I propose: “while the order under natural weathering conditions was:…”

Response 24: According to the suggestion of the Reviewer, sentence have been corrected.

Point 25:  Line 554: replace “degradation” with “analysis” or “storage”

Response 25: Word “degradation” have been replaced with “incubation”.

Point 26:  Line 55: delete “in paraffin”

Response 26: According to the suggestion of the Reviewer, sentence have been corrected.

Point 27:  Line 559: delete the second “paraffin”

Response 27: According to the suggestion of the Reviewer, sentence have been corrected.

Point 28:  Line 561: delete “of aging”

Response 28: According to the suggestion of the Reviewer, sentence have been corrected.

Reviewer 3 Report

This paper is a very interesting study that can contribute to the scope of Materials. The results are well discussed and well organized. Three-dimensional (3D) printing objects with different compositions of polylactide(PLA)/polyhydroxyalkanoate (PHA) were obtained and show potential as a biodegradable container for cosmetic applications. This study demonstrates interesting thermal properties, biodegradable behavior and high dimensional stability under different media conditions and heat up to 55°C. Reviewer cannot find any weak sides of this paper. Doubts are related only with the biocompatibility of these 3D-printed products. Study on cytotoxicity behavior is very important for cosmetic application of 3D printed containers and warrant further analysis/discussion in the manuscript. Finally, reviewer also suggests that biodegradable result is more persuadable to include detail explanation on the dependence of phase miscibility/separation on polymer architecture.

After this small modification the paper should be published.

Author Response

Response to Reviewer 3 Comments

We are indebted for such a positive opinion of the Reviewer regarding our manuscript.

Point 1: Doubts are related only with the biocompatibility of these 3D-printed products. Study on cytotoxicity behavior is very important for cosmetic application of 3D printed containers and warrant further analysis/discussion in the manuscript.

Response 1: Cytotoxicity tests were performed for PLA/PHA blend in the form of dumbbell-shaped specimens obtained from the same filament and were presented in the publication: Gonzalez Ausejo, J.; Rydz, J., et al. A comparative study of three-dimensional printing directions: The degradation and toxicological profile of a PLA/PHA blend. Polym. Degrad. Stab. 2018, 152, 191–207 (reference 12).

Point 2: Finally, reviewer also suggests that biodegradable result is more persuadable to include detail explanation on the dependence of phase miscibility/separation on polymer architecture.

Response 2: The detailed material characteristic including its immiscibility was described also in above publication which is marked in the Materials and Methods section.

Due to the fact that the micrographs did not show significant surface heterogeneity and we did not have a miscible blend for comparison, the effect of phase separation on biodegradation was not discussed separately. In current manuscript, the printing pattern seems to have a more significant effect on biodegradation, which was described in detail in reference 12.

Considerations on the effect of phase separation/miscibility on blend degradation are described in details in the publication: Rydz, J., et al. Forensic engineering of advanced polymeric materials. Part 1 – degradation studies of polylactide blends with atactic poly[(R,S)-3-hydroxybutyrate] in paraffin. Chem. Biochem. Eng. Q. 2015, 29(2), 247–259 (reference 2).

Round  2

Reviewer 2 Report

The manuscript can be published in the present form.